# Proportion of Chinese Children and Adolescents Meeting 24-Hour Movement Guidelines and Associations with Overweight and Obesity

**DOI:** 10.3390/ijerph20021408

**Published:** 2023-01-12

**Authors:** Yi Sun, Yuan Liu, Xiaojian Yin, Ming Li, Ting Zhang, Feng Zhang, Yaru Guo, Pengwei Sun

**Affiliations:** 1College of Physical Education, Ludong University, Yantai 264025, China; 2College of Physical Education & Health, East China Normal University, Shanghai 200241, China; 3College of Economics and Management, Shanghai Institute of Technology, Shanghai 201418, China

**Keywords:** physical activity, screen time, sleep, overweight and obesity, children and adolescents

## Abstract

Background: Since there is little knowledge about the 24-hour movement behaviors of Chinese children and adolescents, the purposes of this study were to investigate the proportion of Chinese children and adolescents meeting the 24-Hour Movement Guidelines and to further evaluate its relationship with overweight and obesity. Methods: A total of 440 children and adolescents aged 7–18 years from 7 cities in China were selected to measure physical activity using accelerometers, and sleep (SLP) and screen time (ST) using questionnaires. The data were analyzed with the independent *T*-test, Mann–Whitney *U* test, Cox–Stuart test, chi-square test, and logistic regression. Results: The proportion of Chinese children and adolescents meeting the overall 24-Hour Movement Guidelines was 7.3%. Boys (11.8%) were higher than girls (3.4%) (*p* < 0.001) and showed a downward trend with age (*P*_trend_ = 0.03). The rates of overweight and obesity among children and adolescents who met the ST, MVPA + ST, ST + SLP, and MVPA + SLP + ST guidelines were 39%, 15%, and 36%, and 25% did not meet any guidelines. The rates of overweight and obesity among those who met 1, 2, and 3 guidelines were lower than the rate among those who did not meet any guidelines (odds ratio (*OR*) = 0.51, 95% confidence interval (*CI*): 0.22–1.17; *OR* = 0.32, 95% *CI*: 0.13–0.77; *OR* = 0.23, 95% *CI*: 0.07–0.81) and showed a decreasing trend (*P*_trend_ = 0.006). Conclusions: The proportion of Chinese children and adolescents meeting the overall 24-Hour Movement Guidelines was low. The rate of overweight and obesity among children and adolescents who met the overall 24-Hour Movement Guidelines was the lowest compared with the rates among those who met any one or two. There was a dose–response relationship between the number of guidelines met and the overweight and obesity rate.

## 1. Introduction

Sleep (SLP), sedentary behavior (SB), and physical activity (PA) form the basis of human behavior, which spans 24 h a day. A large number of studies have shown that these behaviors are closely related to the health of children and adolescents. For example, insufficient SLP, excessive SB, and insufficient PA are important causes of obesity or physical decline in children and adolescents [1,2]. Previous studies have regarded these behaviors as independent health risk factors, pointing to increasing PA as much as possible, reducing SB, and getting adequate SLP. However, some scholars believe that the health effects of a single behavior might be due to the improper use of research methods and that the conclusions were not strictly scientific [3]. In 2016, Tremblay first proposed the concept of “24-hour movement behavior”, which is defined as a continuum of movement from no intensity to high intensity, including SLP, SB, and PA [4]. This concept has fundamentally changed people’s understanding of PA, and the research focus has shifted from the independent effect of a single behavior to the comprehensive effect of combined behaviors. On this basis, Canada took the lead in releasing the first 24-Hour Movement Guidelines for children and adolescents [4]. The guidelines not only stipulate that children and adolescents should have at least 60 min/day of moderate-to-vigorous physical activity (MVPA) but also emphasize that the ST should be less than 2 h/day and appropriate SLP should be ensured (9–11 h for ages 5–13 and 8–10 h for ages 14–17). Many studies have confirmed that children and adolescents meeting the overall 24-Hour Movement Guidelines can bring more health benefits [5,6]. However, the study found that the proportion of children and adolescents worldwide who meet the overall 24-Hour Movement Guidelines is currently low (6.85%) [7]. The compliance of Chinese children and adolescents is facing more severe challenges. Using accelerometers to measure PA, 3 regional studies found that the proportion of children and adolescents in Tianjin was 1.2% [8], 1.0% in Hong Kong [9], and 22.6% in Shanghai. In addition, 5 other studies using questionnaires found that the proportion in China averaged 0.3–7% [10,11,12,13]. Due to the limitations of sample representativeness (regional samples) or research methods (questionnaire surveys), the above data cannot comprehensively and objectively reflect the compliance with the 24-Hour Movement Guidelines in Chinese children and adolescents.

In the past decades, the rate of overweight and obesity among children and adolescents worldwide has continued to rise. The resulting health problems and potential health threats have become important public health problems to be solved urgently. From 1989 to 2014, the overweight and obesity rate in Chinese children and adolescents soared from 1.1% to 20.4% [14]. The latest data in 2019 show that the rising rate of overweight and obesity among children and adolescents has not been reversed, and there is an urgent need to take effective intervention measures to change this situation [15]. In recent years, more and more studies have shown that the compliance with the 24-Hour Movement Guidelines of children and adolescents is closely related to overweight and obesity. Studies in Hong Kong [9], the Czech Republic [16], and Japan [17] found that meeting the two guidelines can significantly reduce the rate of overweight and obesity. Many other studies have found that children and adolescents who meet the three guidelines have the lowest overweight and obesity rate or BMI-Z [8,17,18,19,20,21]. Recently, Scott et al. [6] and Wang [22] systematically reviewed the relevant research in recent years. These studies found that meeting the 24-Hour Movement Guidelines was negatively correlated with all obesity indicators (BMI, waist circumference, and fat mass). In addition, other studies have found that there is a significant dose–response relationship between the number of guidelines met and the rate of overweight and obesity [19,23,24]. However, there are few relevant studies in the Chinese Mainland, and the conclusions are inconsistent. Wang et al. [12] found that meeting the overall 24-Hour Movement Guidelines is not associated with overweight and obesity but that adequate SLP can help reduce the rate of overweight and obesity in children and adolescents. Chen et al. [10] showed a different finding in their study, which indicated that children and adolescents who met the overall 24-Hour Movement Guidelines or a combination of the MVPA and SLP guidelines had a lower rate of overweight and obesity. The above two studies may have a large bias in using questionnaires to investigate PA, and the researchers also expressed the same concern regarding the limitation [10,12]. Therefore, more objective methods are needed to accurately evaluate the associations between compliance with the 24-Hour Movement Guidelines and overweight and obesity in Chinese children and adolescents.

To sum up, the purposes of this study were as follows: (1) to investigate the proportion of Chinese children and adolescents meeting the 24-Hour Movement Guidelines; (2) to evaluate the associations between the 24-Hour Movement Guidelines met and overweight and obesity.

## 2. Materials and Methods

### 2.1. Participants

From March to July 2019, the research team recruited 840 children and adolescents aged 7–18 in 7 cities for PA measurement and the questionnaire survey: Shanghai (East China), Taiyuan (North China), Guangzhou (South China), Changsha (Central China), Urumqi (Northwest), and Chengdu and Kunming (Southwest). From each city, 1 primary school, 1 junior high school, and 1 senior high school were randomly selected, and 5 boys and 5 girls from each grade were randomly selected from each school, totaling 120 children and adolescents. Inclusion criteria: children and adolescents aged 7–18 who can perform physical activities normally. Exclusion criteria: children and adolescents that have diseases that can lead to a poor nutritional status (wasting, overweight, or obesity) or are taking medications that can lead to changes in energy metabolism. Before the investigation, the participants and their parents were told the purpose of the study. Written informed consent was obtained from both the participants and their parents, and the investigation was conducted as an anonymous survey. This study was approved by the Human Experimental Ethics Committee of East China Normal University (no. HR 006-2019). After the validity test of the accelerometer data, 119 participants were excluded. The validity test criteria for accelerometer wearing will be described in detail later. In addition, 281 participants were excluded because they did not fill in the ST or SLP items in the questionnaire. This study finally included 440 eligible participants, comprising 203 boys and 237 girls. Figure 1 showed the process of participants exclusion. After testing, there was no significant difference between excluded and included participants in age (*p* = 0.74) and overweight and obesity rate (*p* = 0.20), but the proportion of boys in excluded participants was higher (54.2%, *p* = 0.03).

### 2.2. Physical Activity

Physical activity was measured using the ActiGraph GT3X+ accelerometer (ActiGraph, Pensacola, FL, USA). During measurement, the accelerometer was worn on the right wrists of the participants for 7 consecutive days (including 5 school days and 2 weekends). The epoch duration was set to 5 s. An accelerometer wearing time of more than 600 min per day was defined as a valid day, and the accelerometer wearing criteria including 3 valid days or more and at least 2 school days and 1 weekend (Saturday and/or Sunday) were included [25]. The cut-points of Evenson [26] were adopted to classify SB (0–100 counts/min), light physical activity (101–2295 counts/min), moderate physical activity (2296–4011 counts/min), and vigorous physical activity (4012 counts/min or more). The cut-points used in this study had high validity and reliability in evaluating the PA of children and adolescents [27].

### 2.3. Screen Time and Sleep

A questionnaire was used to evaluate ST and SLP time. Participants aged 7–9 were assisted by their parents to fill out the questionnaire, and participants aged 10–18 filled out the questionnaire themselves. The participants reported their typical time spent watching TV/movies, using computers, and playing on mobile phones/tablets per weekday and per weekend day. The participants also reported their typical time to go to bed at night and the time to get up in the morning per weekday, and per weekend day, taking the difference as the SLP time. The ST and SLP time are weighted averages according to weekdays (5/7) and weekend days (2/7).

### 2.4. Ascertainment of Overweight and Obesity

The height and weight of the participants were measured using a standard procedure with an accuracy of 0.1 cm and 0.1 kg, respectively. BMI was calculated. In this study, participants were divided into normal weight, overweight, and obesity according to the standards formulated by the Group of China Task Force [28].

### 2.5. Compliance with 24-Hour Movement Guidelines

Compliance was judged according to the Canadian 24-Hour Movement Guidelines for Children and Youth [4]. The specifics guidelines are as follows: MVPA ≥ 60 min/day, ST < 2 h/day, and SLP time of 9–11 h/day (for children) or 8–10 h/day (for adolescents).

### 2.6. Covariates

Socioeconomic status (SES) is closely associated with overweight and obesity in children and adolescents, so it was considered as a potential confounder. The research team investigated the participants’ parents’ educations, occupations, and family incomes through a questionnaire. The specific calculation method of SES has been described in other studies [29]. In brief, the method was as follows: (1) assigning points to parents’ educations (according to years of education), occupations (according to international occupational standard classification), and family incomes (below 2000, 2 points; 2001–5000, 5 points; 5001–8000, 8 points; and 8000+, 10 points); (2) screening or conversion; (3) processing missing values; and (4) converting all variables into standard scores and performing a principal component analysis to obtain SESs.

### 2.7. Statistical Analysis

All statistical analyses were conducted using SPSS 25.0 (IBM Company, Armonk, NY, USA). This study used a histogram, P–P diagram, and Q–Q diagram to comprehensively judge the normality of the data. Normal data were expressed as means ± SD, and skewed data were expressed as medians (25th, 75th percentiles). The differences between sex were analyzed with the independent *T*-test and with the Mann–Whitney *U* test for continuous variables and the chi-square test for categorical variables. The Cox–Stuart test was used to judge the change trend of compliance with the 24-Hour Movement Guidelines with age. A binary logistic regression analysis was used to analyze the odds of overweight and obesity in participants who did not meet any guidelines, met the specified or any one guideline, met the specified or any two guidelines, and met the overall guidelines, and a trend test was conducted. The level of significance was set at *p* < 0.05.

## 3. Results

Table 1 shows that the MVPA, SLP, and overweight and obesity rates of boys are higher than those of girls, and the difference is statistically significant (*p* < 0.05). There was no significant sex difference in ST (*p* = 0.11).

Figure 2 shows the proportion of Chinese children and adolescents meeting the 24-Hour Movement Guidelines. The proportions of Chinese children and adolescents meeting the MVPA, ST, SLP, or all 3 guidelines and that of those not meeting any guidelines were 17.3%, 69.1%, 56.2%, 7.3%, and 11.8%, respectively.

Table 2 shows that there was a significant sex difference in compliance with the 24-Hour Movement Guidelines among Chinese children and adolescents (*χ*^2^ = 24.8, *p* < 0.001). Among them, the proportion of boys (5.9%) not meeting any guideline was lower than that of girls (16.9%); the proportion of boys (11.8%) meeting the 3 guidelines was higher than that of girls (3.4%).

Figure 3 shows the proportions of Chinese children and adolescents aged 7–18 meeting 0, 1, 2, or 3 of the 24-Hour Movement Guidelines. The proportion of Chinese children and adolescents meeting 3 and 2 guidelines gradually decreases with age (*P*_trend_ = 0.03), and the proportion meeting 1 guideline and not meeting any guidelines gradually increases with age (*P*_trend_ = 0.03). Fifteen years old is the inflection point of compliance with guidelines. From the age of 15, the proportions meeting 3 guidelines, 2 guidelines, and not meeting any guidelines decrease suddenly, while the proportion meeting 1 guideline increased significantly. (See Table A1 for detailed data.)

Table 3 shows that compared with children and adolescents who did not meet any guidelines, the risk of meeting ST was 39% (*OR* = 0.39, 95% *CI* =0.15–0.99), the risk of meeting MVPA and ST was 15% (*OR* = 0.15, 95% *CI* =0.03–0.78), the risk of meeting ST and SLP was 36% (*OR* = 0.36, 95% *CI* =0.15–0.89), and the risk of meeting all three was 25% (*OR* = 0.25, 95% *CI* =0.07–0.87).

Table 4 shows that as the number of guidelines met increased, the risk of overweight and obesity decreased in children and adolescents (meeting 1 guideline: *OR* = 0.51, 95% *CI*: 0.22–1.17; meeting 2 guidelines: *OR* = 0.32, 95% *CI*: 0.13–0.77; meeting 3 guidelines: *OR* = 0.23, 95% *CI*: 0.07–0.81; *P*_trend_ = 0.006). After splitting by sex, the same level of significance was found in boys but not in girls.

## 4. Discussion

In this study, a nationwide representative sample was selected, physical activity was measured using an accelerometer, ST and SLP time were measured via a questionnaire, and compliance with the 24-Hour Movement Guidelines in Chinese children and adolescents aged 7–18 was objectively investigated, and the relationship between compliance with the guidelines and overweight and obesity was further evaluated. The study found that the proportion of Chinese children and adolescents meeting the overall 24-Hour Movement Guidelines was low. The rate of overweight and obesity among children and adolescents who met the overall 24-Hour Movement Guidelines was the lowest compared with those who met any one or two. There was a dose–response relationship between the number of guidelines met and the overweight and obesity rate.

In order to formulate more targeted public health policies, it is necessary to comprehensively understand the characteristics of the 24-Hour Movement Guidelines for children and adolescents in terms of compliance, sex, and age [30]. In terms of compliance, the proportion of Chinese children and adolescents aged 7–18 who meet the overall 24-Hour Movement Guidelines is about 7.3%, which is slightly higher than the world average (7.12%) [7], far lower than Canada (17.5%) [31] and Australia (14.9%) [8], slightly lower than the United States (8.8%) [32] and Japan (10.5%) [17], and higher than South Korea (1.6%) [33]. Specifically, the proportions meeting the ST (69.1%) and SLP (56.2%) guidelines in Chinese children and adolescents are high. However, due to the different investigation methods for ST and SLP, the compliance with the ST and SLP guidelines in children and adolescents in different countries vary greatly, which needs to be carefully compared horizontally. Although different studies have great differences in ST and SLP, a common denominator found in most studies is the low compliance with the MVPA guideline, which is the main reason for the low overall compliance with the 24-Hour Movement Guidelines [9,21]. Low compliance with the MVPA guideline is an important problem in children and adolescents worldwide. Guthold et al. [34] analyzed 298 population studies from 2001 to 2016 and found that the average MVPA compliance rate in global children and adolescents in 2016 was only 19.0%, which was slightly higher than 17.4% in 2001. Therefore, increasing MVPA is the key to improving the overall compliance of children and adolescents. In terms of sex differences, the overall compliance rate in Chinese boys (11.8%) is higher than that in girls (3.4%), which may be mainly due to the higher compliance rate for MVPA in boys. In this study, the compliance rate for MVPA in boys was 29.6%, much higher than 6.8% in girls. Sex differences in overall compliance rates were not found in the Spanish and American studies and may also be related. The MVPA compliance rate in boys in Spain [35] and the United States [32] is higher than that in girls, but the range of sex difference is much smaller than that in China (Spain: 15.6%; the United States: 7.1%). In terms of age difference, the overall compliance rate for the 24-Hour Movement Guidelines in Chinese children and adolescents shows a downward trend with age. Friel et al. [32] also found the same conclusion in the investigation of children and adolescents aged 6–17 in the United States. The researchers found that from the ages of 6 to 17, the overall compliance rate in American children and adolescents decreased by 13%. In addition, the age trend of Chinese children and adolescents has another important feature: 15 years old is the inflection point of compliance with guidelines. From the age of 15, the proportions meeting 3 guidelines, 2 guidelines, and not meeting any guideline all decrease suddenly, while the proportion meeting 1 guideline increases significantly. The reason may be related to the fact that Chinese children and adolescents are at the key time point of the high school entrance examination at the age of 15. At this stage, the academic pressure on Chinese teenagers is high, and MVPA, ST, and SLP are reduced, resulting in a decrease in the MVPA and SLP compliance rates, while the ST compliance rate increases (see Table A2 for detailed data).

Long-term ST is one of the important risk factors for overweight and obesity in children and adolescents. The meta-analysis of Fang et al. [36] showed that children and adolescents with an ST greater than 2 h/d had a 1.67 times higher risk of overweight and obesity than those with an ST of less than 2 h/d. Wu et al. [37] further conducted a quantitative study. The results showed that the BMI of children and adolescents with an ST greater than 2 h/d was 0.7 kg/m^2^ higher than that of children and adolescents with an ST of less than 2 h/d, and the ST of obese children and adolescents was 0.3 h/d higher than that of normal children. Previous studies have found that meeting ST and its combination was related to lower BMI, FM%, and visceral fat [16,38], and the rate of overweight and obesity was lower [10]. Consistent with previous studies, this study found that the rates of overweight and obesity among children and adolescents meeting the ST and ST guideline combinations (ST + MVPA, ST + SLP, and ST + MVPA + SLP) were relatively low. A further analysis of covariance (controlling age, sex, SES, and MVPA) was conducted in this study, and the results were close to those of Wu et al. The results showed that compared with children and adolescents who met the ST guideline, the BMI of those who did not meet the ST guideline was 0.85 kg/m^2^ higher (*F* = 8.84, *p* = 0.003). Compared with normal--weight children and adolescents, the ST of overweight and obese children and adolescents was 11.3 min/d higher (*F* = 0.84, *p* = 0.36). (See Table A3 and Table A4 for detailed data.) Although reducing ST is associated with a statistically significant reduction in overweight and obesity rates, and reducing ST is relatively easier than increasing MVPA and SLP, aiming only for a reduction in ST within interventions could be a limited strategy in ameliorating excessive ST in children and adolescents [39] because ST is associated with other lifestyle choices (such as PA, SLP, and diet), which interact to promote obesity [40]. However, this study did not find that meeting the MVPA and SLP guidelines and their combination (MVPA + SLP) were significantly associated with overweight and obesity. This result is not unexpected. Some studies have not found a significant association between MVPA [10,41,42] and SLP [43,44,45] and overweight and obesity in children and adolescents. Although this study and the above studies have not found a significant cross-sectional correlation, there is evidence that a lack of MVPA and SLP may have adverse effects on the long-term health of children and adolescents [46,47].

A study of the dose–response relationship can quantitatively clarify the changing rule between compliance with the 24-Hour Movement Guidelines and overweight and obesity. Laurson et al. [23] examined the dose–response relationship between the number of guidelines met and the obesity rate in children aged 7–12. The study found that children who met 2, 1, and none of the guidelines had a 2.6-, 4.7-, and 8.2-fold higher risk of obesity than those who met all 3 guidelines. Carson et al. [3], Janssen et al. [23], and Katzmarzyk et al. [19] further examined the dose–response relationship between BMI, waist circumference, fat mass, subcutaneous fat, and abdominal fat and the number of guidelines met. The results showed that all obesity indicators decreased linearly with an increase in the number of guidelines met. Consistent with previous studies, this study found that there was a dose–response relationship between the number of guidelines met and the overweight and obesity rate in children and adolescents, that is, with an increase in the number of guidelines met, the overweight and obesity rate decreased linearly. This conclusion has important reference significance for government departments to formulate public health strategies. Although this study did not find significant health benefits when examining a certain behavior alone (such as MVPA and SLP), there was a significant superposition effect when comprehensively investigating the health benefits of 24-h activity behavior. Compared with the promotion policy of a single behavior (MVPA, SLP, or ST), the findings of this study reinforce the importance of behavior combination strategies in preventing overweight and obesity in children and adolescents. The data of this study showed that children and adolescents who did not meet any guidelines had the highest risk of being overweight and obese (see Table A5 for detailed data). Therefore, it may be more important to consider the comprehensive effects of MVPA, SLP, and ST on health from the overall perspective, and children and adolescents should be promoted to meet the guidelines as much as possible. However, studies in Hong Kong [9] and the Czech Republic [16] did not find a significant association between meeting the overall guidelines and overweight and obesity. The reason may be that the proportion of children and adolescents in the above two regions meeting the overall guidelines is too low and the sample size is not large, resulting in unreliable results of the regression analyses [16]. Similarly, in the study of preschool children, some studies also did not find that adherence to any single recommendation or any combination of recommendations was associated with being overweight and obese [48,49]. In addition, it is important to note that this study only found a significant association between the overall sample and boys, which may also be related to the low compliance rate in girls. Of course, it cannot be ruled out that the 24-h movement behavior of girls may not be related to overweight and obesity, because the weight status of girls may be more susceptible to dietary, environmental, and cultural factors [50].

The strength of this study was its use of a representative sample of Chinese children and adolescents from seven cities in six administrative regions. Moreover, this is the first study to adopt accelerometers to investigate compliance with the 24-Hour Movement Guidelines for Chinese children and adolescents aged 7–18. This study also has some limitations in its research methods. First, a high proportion of the 400 participants were excluded from this study due to an insufficient wearing time of the accelerometer or unqualified filling of the questionnaire. However, through a comparison between the included and excluded samples, it was not found that the excluded samples had a significant impact on the key indicators. Second, the questionnaire used in this study to evaluate SLP may be biased, and a more objective method should be adopted in the future. Third, this study included 26 (5.9%) 18-year-old adolescents, but the Canadian 24-Hour Movement Guidelines for Children and Youth are only applicable to children and adolescents aged 5–17 years. However, according to the WHO’s age definition of adolescents (10–19 years old), there is reason for this study to classify the 18-year-old group as adolescents [9].

## 5. Conclusions

The proportion of Chinese children and adolescents meeting the overall 24-Hour Movement Guidelines was low. The rate of overweight and obesity among children and adolescents who met the overall 24-Hour Movement Guidelines was the lowest compared with those who met any one or two. There was a dose–response relationship between the number of guidelines met and the overweight and obesity rate.

## Figures and Tables

**Figure 1 ijerph-20-01408-f001:**
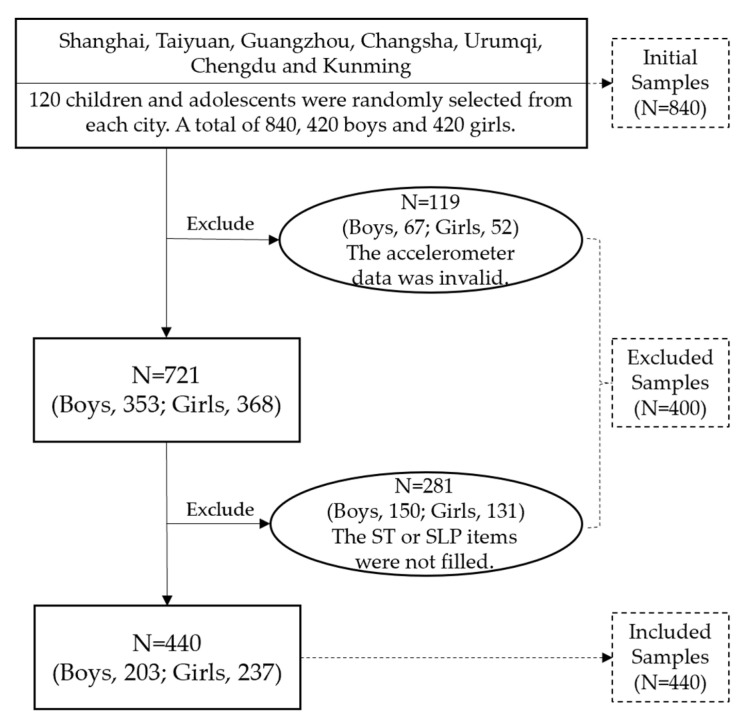
Flow chart of sample inclusion.

**Figure 2 ijerph-20-01408-f002:**
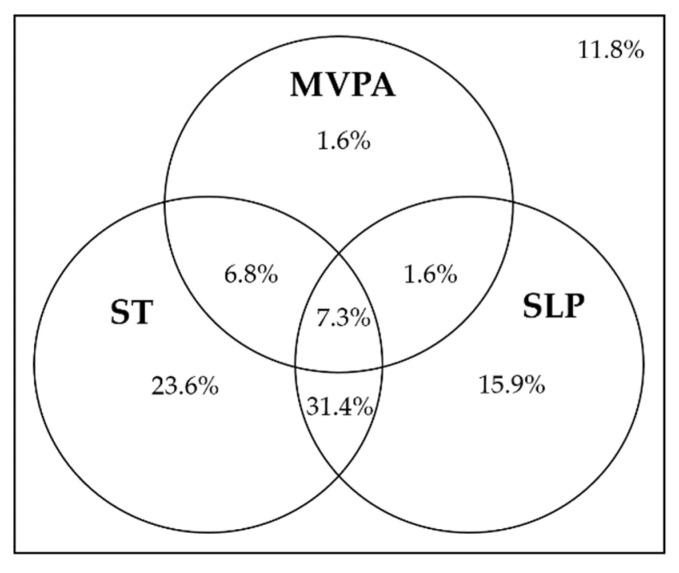
Proportions of Chinese children and adolescents meeting 24-Hour Movement Guidelines.

**Figure 3 ijerph-20-01408-f003:**
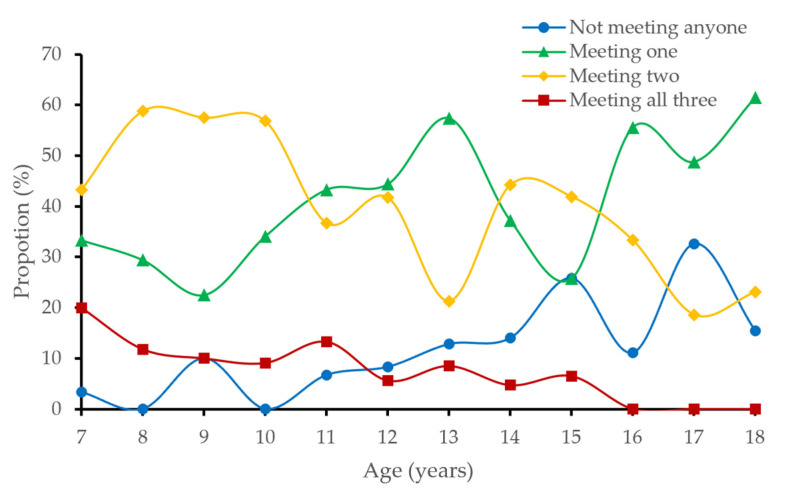
Proportions of Chinese children and adolescents aged 7–18 meeting 0, 1, 2, or 3 of the 24-Hour Movement Guidelines.

**Table 1 ijerph-20-01408-t001:** Descriptive characteristics of participants.

	Total (*n* = 440)	Boys (*n* = 203)	Girls (*n* = 237)	*p*-Value
Age (years)	12.5 ± 3.3	12.4 ± 3.3	12.6 ± 3.3	0.60
Height (cm)	151.7 ± 17.5	153.5 ± 19.5	150.1 ± 15.4	0.04
Weight (kg)	43.4 ± 14.5	45.4 ± 16.7	41.7 ± 12.2	0.009
BMI (kg/m^2^)	18.3 ± 3.0	18.6 ± 3.2	18.1 ± 2.9	0.07
SES	0.04 ± 1.03	0.08 ± 1.11	−0.01 ± 0.96	0.38
SB (min/d)	565.3 ± 121.2	542.2 ± 122.4	585.1 ± 116.9	<0.001
LPA (min/d)	268.2 ± 86.5	278.7 ± 89.4	259.2 ± 83.1	0.02
MVPA (min/d)	41.3 ± 20.1	49.3 ± 21.3	34.4 ± 16.2	<0.001
SLP (h/d)	8.2 ± 1.3	8.4 ± 1.3	8.1 ± 1.3	0.03
ST (min/d)	97.2 (34.4, 166.1)	101.1 (46.8, 172.5)	92.9 (32.1, 165.3)	0.11
Weight status	*n* (%)	*n* (%)	*n* (%)	
Normal weight	359 (81.6)	152 (74.9)	207 (87.3)	0.04
Overweight and obesity	81 (18.4)	51 (25.1)	30 (12.7)	

Note: values are presented as means ± SD and medians (25th, 75th percentiles); BMI, body mass index; SES, socioeconomic status; SB, sedentary behavior; LPA, light physical activity; MVPA, moderate-to-vigorous physical activity; SLP, sleep; ST, screen time.

**Table 2 ijerph-20-01408-t002:** Sex differences in compliance with 24-Hour Movement Guidelines among Chinese children and adolescents.

The Number of Guidelines Met	Boys	Girls	*χ* ^2^	*p*-Value
0	12 (5.9%)	40 (16.9%)	24.8	<0.001
1	77 (37.9%)	104 (43.9%)		
2	90 (44.3%)	85 (35.9)		
3	24 (11.8%)	8 (3.4%)		

**Table 3 ijerph-20-01408-t003:** Associations between meeting the 24-Hour Movement Guidelines and overweight and obesity in Chinese children and adolescents.

Guidelines	Total (*n* = 440) ^a^		Boys (*n* = 203) ^b^		Girls (*n* = 237) ^b^	
	*OR*	95% *CI*	*OR*	95% *CI*	*OR*	95% *CI*
None	Reference		Reference		Reference	
MVPA only	0.38	0.04~3.77	0.20	0.02~2.37	NA	
ST only	0.39	0.15~0.99	0.30	0.08~1.22	0.30	0.07~1.29
SLP only	0.73	0.29~1.85	0.34	0.08~1.41	1.16	0.30~4.50
MVPA + ST only	0.15	0.03~0.78	0.10	0.02~0.63	NA	
MVPA + SLP only	0.56	0.09~3.56	0.30	0.04~2.31	NA	
ST + SLP only	0.36	0.15~0.89	0.17	0.04~0.69	0.50	0.14~1.82
All three	0.25	0.07~0.87	0.11	0.02~0.57	0.87	0.11~6.83

Note: ^a^, adjusted for sex, age, and SES; ^b^, adjusted for age and SES. NA, not applicable, indicating insufficient sample size for estimation.

**Table 4 ijerph-20-01408-t004:** Associations between the number of guidelines met and overweight and obesity in Chinese children and adolescents.

The Number of Guidelines Met	Total (*n* = 440) ^a^		Boys (*n* = 203) ^b^		Girls (*n* = 237) ^b^	
	*OR*	95% *CI*	*OR*	95% *CI*	*OR*	95% *CI*
0	Reference		Reference		Reference	
1	0.51	0.22–1.17	0.31	0.08–1.11	0.56	0.17–1.89
2	0.32	0.13–0.77	0.16	0.04–0.60	0.46	0.13–1.63
3	0.23	0.07–0.81	0.10	0.02–0.53	0.83	0.11–6.42
*P* _trend_	0.006		0.002		0.45	

Note: ^a^, adjusted for sex, age, and SES; ^b^, adjusted for age and SES.

## Data Availability

The data that support the findings of this study are available from the corresponding author upon reasonable request.

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
