# Peer review of "Proportion of Chinese Children and Adolescents Meeting 24-Hour Movement Guidelines and Associations with Overweight and Obesity"

_ijerph, 2023, doi:10.3390/ijerph20021408_

Round 1

Reviewer 1 Report

This is an observational study evaluating the proportion of Chinese children and adolescents meeting 24-hour movement guidelines and the association with overweight and obesity. Although potentially useful and interesting, there are several concerns about the methodology that do not make the validity of the study possible.

Methods - Participants - Page 3 - Line 100-117:

-The population of origin is not clearly defined. The authors say they "recruited 840 children and adolescents... (line 101)" and that 440 ultimately enrolled in their sample as eligible participants (line 113).

-Selection criteria were probably applied in this methodology that we do not know, since 119 people were excluded after the accelerometer data validity test (line 111) and 281 people were excluded when filling out the questionnaire (line 112), so both samples of people excluded would be part of your result.

-The authors must clarify what the initial sample is, the population that remains after applying the selection criteria and finally, the sample that is analyzed, considering as dropouts those who start the study but for some reason do not finish it.

Reviewer 2 Report

Very needed research in our modern society. It showed that the proportion of Chinese children and adolescents meeting the overall 24-Hour Movement Guidelines was low. For those who met the overall 24-Hour Movement Guidelines the rate of overweight and obesity was the lowest. There was a dose-response relationship between the number of guidelines met and the overweight and obesity rate. It clearly show that people need some strict guidelines to keep proper body composition. This type of research is not new and there are many similar papers, but the problem is so important, that I recommend this paper to be published in the journal. English level is acceptable and the composition of the paper is proper.

Some papers could be added to the Introduction and/or Discussion.

Santos, R., Zhang, Z., Pereira, J.R. et al. Compliance with the Australian 24-hour movement guidelines for the early years: associations with weight status. BMC Public Health 17 (Suppl 5), 867 (2017). https://doi.org/10.1186/s12889-017-4857-8

The sample comprised 202 toddlers (104 girls) aged 19.74 ± 4.07 months from the GET UP! Study. Participants wore accelerometers (Actigraph GT3X+) for 24 h over 7 consecutive days to assess physical activity, sedentary time and sleep. Parents reported participants’ screen time. Weight and height were measured and body mass index (BMI) z-scores by age and sex were calculated. Analysis of Covariance (ANCOVA) was performed to test differences in BMI z-scores between participants complying with (i) none or any individual guideline, (ii) any combination of meeting two guidelines, and (iii) those who met all three guidelines, adjusting for child age, gender and socioeconomic status.

Only 8.9% of the sample met the overall 24 h movement guidelines. Most of the sample met the physical activity (96.5%) and sleep (79.7%) guidelines but only 11.4% met the sedentary behavior guideline. Average BMI Z-scores did not significantly differ between children who complied with none or any individual guideline, any combination of meeting two guidelines, and those who met all three guidelines (p > 0.05). Although the lack of significant differences, participants who accomplished any combination of two guidelines or all three guidelines appear to have had a lower BMI Z-score than those complying with one of the guidelines or none.

Just under 9% of our sample met the overall Australian 24 h Movement Guidelines for the Early Years. BMI was not associated with the accomplishment of any of the 24-h Movement Guidelines. Strategies to promote adherence to the 24-h movement guidelines in toddlers, particularly for screen time, are necessary, as promoting health-related behaviors in early childhood has the potential to provide children a strong foundation for lifelong physical and mental health.

Berglind D, Ljung R, Tynelius P, Brooke HL. Cross-sectional and prospective associations of meeting 24-h movement guidelines with overweight and obesity in preschool children. Pediatr Obes. 2018 Jul;13(7):442-449. doi: 10.1111/ijpo.12265. Cross-sectional studies report that meeting the newly developed 24-h movement guidelines (≥60 min moderate to vigorous physical activity (MVPA), ≤120 min screen time and 9-11 h sleep duration) are associated with lower adiposity indicators in children. However, prospective data are absent.

The study sample consisted of 830 children from the PRIMROSE study with GT3X+ accelerometer measured physical activity and parent reported screen time and sleep duration at age 4 years and objectively measured anthropometrics at age 4 and 5 years. The main outcome variables were weight status, body mass index (BMI) and BMI z-score at ages 4 and 5 years. Exposure variables were defined as meeting vs. not meeting the 24-h movement guidelines and combinations of these recommendations.

On average, 18.4% of the total study sample met the combination of MVPA, sleep duration and screen time recommendations. In isolation, the MVPA, screen time and sleep guidelines were met by 31%, 63% and 98% of the total study sample, respectively. Adherence to any single recommendation, or any combination of recommendations at age 4 years, was not associated with being overweight or obese nor with BMI and BMI z-score at age 4 or 5 years.

In contrast to previous cross-sectional studies, neither individual movement behaviours nor combinations of behaviours at age 4 years was associated with overweight or obesity, BMI or BMI z-score at age 4 or 5 years.

Author Response

Dear Reviewer:

On behalf of my co-authors, we are very grateful to you for giving us an opportunity to revise our manuscript. We appreciate you very much for your positive and constructive comment on our manuscript. We have studied your comment carefully and tried our best to revise our manuscript. If there are any other modifications we could make, we would like to modify them, and we really appreciate your help. The following is my response to your comment. Thanks again to the hard work of you.

Comment No. 1: Some papers could be added to the Introduction and/or Discussion.

Response: Thank you for your comment. In the discussion, we mentioned the relationship between preschool children's adherence to 24-Hour Movement Guidelines and overweight and obesity, and cited two articles you provided. Through the above modifications, we can understand the relationship between adherence to 24-Hour Movement Guidelines and overweight and obesity in a more comprehensive age group. The specific modifications are as follows: “Similarly, in the study of preschool children, some studies also did not find that adherence to any single recommendation or any combination of recommendations was associated with being overweight and obesity [48-49]. (Line 327-329)”

Round 2

Reviewer 1 Report

I have rereviewed the manuscript entitled “Proportion of Chinese Children and Adolescents Meeting 24-Hour Movement Guidelines and Association with Overweight and Obesity” for consideration to be published in International Journal of Environmental Research and Public Health. It is a very interesting study that probably will encourage another authors to perform further studies. Congratulations to the authors.

Author Response

Dear reviewer:

Thank you very much for your constructive comments on our work.